# Failing Loudly: An Empirical Study of Methods for Detecting Dataset Shift

**Stephan Rabanser**[*]
AWS AI Labs
rabans@amazon.com

**Stephan Günnemann**
Technical University of Munich
guennemann@in.tum.de

**Zachary C. Lipton**
Carnegie Mellon University
zlipton@cmu.edu

## Abstract

We might hope that when faced with unexpected inputs, well-designed software systems would fire off warnings. Machine learning (ML) systems, however, which depend strongly on properties of their inputs (e.g. the i.i.d. assumption), tend to fail silently. This paper explores the problem of building ML systems that fail loudly, investigating methods for detecting dataset shift, identifying exemplars that most typify the shift, and quantifying shift malignancy. We focus on several datasets and various perturbations to both covariates and label distributions with varying magnitudes and fractions of data affected. Interestingly, we show that across the dataset shifts that we explore, a two-sample-testing-based approach, using pre-trained classifiers for dimensionality reduction, performs best. Moreover, we demonstrate that domain-discriminating approaches tend to be helpful for characterizing shifts qualitatively and determining if they are harmful.

## 1 Introduction

Software systems employing deep neural networks are now applied widely in industry, powering the vision systems in social networks [47] and self-driving cars [5], providing assistance to radiologists [24], underpinning recommendation engines used by online platforms [9, 12], enabling the best-performing commercial speech recognition software [14, 21], and automating translation between languages [50]. In each of these systems, predictive models are integrated into conventional human-interacting software systems, leveraging their predictions to drive consequential decisions.

The reliable functioning of software depends crucially on tests. Many classic software bugs can be caught when software is compiled, e.g. that a function receives input of the wrong type, while other problems are detected only at run-time, triggering warnings or exceptions. In the worst case, if the errors are never caught, software may behave incorrectly without alerting anyone to the problem.

Unfortunately, software systems based on machine learning are notoriously hard to test and maintain [42]. Despite their power, modern machine learning models are brittle. Seemingly subtle changes in the data distribution can destroy the performance of otherwise state-of-the-art classifiers, a phenomenon exemplified by adversarial examples [51, 57]. When decisions are made under uncertainty, even shifts in the label distribution can significantly compromise accuracy [29, 56]. Unfortunately, in practice, ML pipelines rarely inspect incoming data for signs of distribution shift. Moreover, best practices for detecting shift in high-dimensional real-world data have not yet been established[2].

In this paper, we investigate methods for detecting and characterizing distribution shift, with the hope of removing a critical stumbling block obstructing the safe and responsible deployment of machine learning in high-stakes applications. Faced with distribution shift, our goals are three-fold:

---

[*]Work done while a Visiting Research Scholar at Carnegie Mellon University.

[2]TensorFlow's data validation tools compare only summary statistics of source vs target data: https://tensorflow.org/tfx/data_validation/get_started#checking_data_skew_and_drift

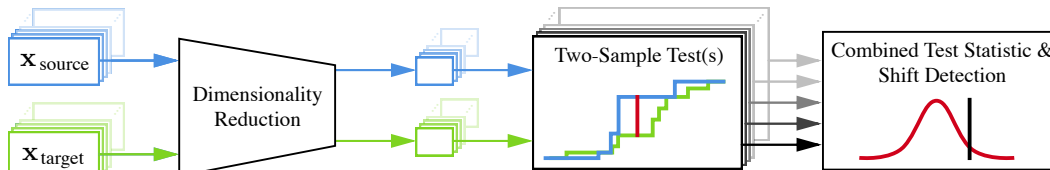

Figure 1: Our pipeline for detecting dataset shift. Source and target data is fed through a dimensionality reduction process and subsequently analyzed via statistical hypothesis testing. We consider various choices for how to represent the data and how to perform two-sample tests.

(i) detect when distribution shift occurs from as few examples as possible; (ii) characterize the shift, e.g. by identifying those samples from the test set that appear over-represented in the target data; and (iii) provide some guidance on whether the shift is harmful or not. As part of this paper we principally focus on goal (i) and explore preliminary approaches to (ii) and (iii).

We investigate shift detection through the lens of statistical two-sample testing. We wish to test the equivalence of the *source* distribution $p$ (from which training data is sampled) and *target* distribution $q$ (from which real-world data is sampled). For simple univariate distributions, such hypothesis testing is a mature science. However, best practices for two sample tests with high-dimensional (e.g. image) data remain an open question. While off-the-shelf methods for kernel-based multivariate two-sample tests are appealing, they scale badly with dataset size and their statistical power is known to decay badly with high ambient dimension [37].

Recently, Lipton et al. [29] presented results for a method called *black box shift detection (BBSD)*, showing that if one possesses an off-the-shelf label classifier $f$ with an invertible confusion matrix, then detecting that the source distribution $p$ differs from the target distribution $q$ requires only detecting that $p(f(\boldsymbol{x})) \neq q(f(\boldsymbol{x}))$. Building on their idea of combining black-box dimensionality reduction with subsequent two-sample testing, we explore a range of dimensionality-reduction techniques and compare them under a wide variety of shifts (Figure 1 illustrates our general framework). We show (empirically) that BBSD works surprisingly well under a broad set of shifts, even when the label shift assumption is not met. Furthermore, we provide an empirical analysis on the performance of domain-discriminating classifier-based approaches (i.e. classifiers explicitly trained to discriminate between source and target samples), which has so far not been characterized for the complex high-dimensional data distributions on which modern machine learning is routinely deployed.

## 2   Related work

Given just one example from the test data, our problem simplifies to *anomaly detection*, surveyed thoroughly by Chandola et al. [8] and Markou and Singh [33]. Popular approaches to anomaly detection include density estimation [6], margin-based approaches such as the one-class SVM [40], and the tree-based isolation forest method due to [30]. Recently, also GANs have been explored for this task [39]. Given simple streams of data arriving in a time-dependent fashion where the signal is piece-wise stationary with abrupt changes, this is the classic time series problem of change point detection, surveyed comprehensively by Truong et al. [52]. An extensive literature addresses dataset shift in the context of domain adaptation. Owing to the impossibility of correcting for shift absent assumptions [3], these papers often assume either covariate shift $q(\boldsymbol{x}, y) = q(\boldsymbol{x})p(y|\boldsymbol{x})$ [15, 45, 49] or label shift $q(\boldsymbol{x}, y) = q(y)p(\boldsymbol{x}|y)$ [7, 29, 38, 48, 56]. Schölkopf et al. [41] provides a unifying view of these shifts, associating assumed invariances with the corresponding causal assumptions.

Several recent papers have proposed outlier detection mechanisms dubbing the task *out-of-distribution (OOD) sample detection*. Hendrycks and Gimpel [19] proposes to threshold the maximum softmax entry of a neural network classifier which already contains a relevant signal. Liang et al. [28] and Lee et al. [26] extend this idea by either adding temperature scaling and adversarial-like perturbations on the input or by explicitly adapting the loss to aid OOD detection. Choi and Jang [10] and Shalev et al. [44] employ model ensembling to further improve detection reliability. Alemi et al. [2] motivate use of the variational information bottleneck. Hendrycks et al. [20] expose the model to OOD samples, exploring heuristics for discriminating between in-distribution and out-of-distribution samples. Shafaei et al. [43] survey numerous OOD detection techniques.

# 3   Shift Detection Techniques

Given labeled data $\{(\boldsymbol{x}_1, y_1), ..., (\boldsymbol{x}_n, y_n)\} \sim p$ and unlabeled data $\{\boldsymbol{x}_1', ..., \boldsymbol{x}_m'\} \sim q$, our task is to determine whether $p(\boldsymbol{x})$ equals $q(\boldsymbol{x}')$. Formally, $H_0 : p(\boldsymbol{x}) = q(\boldsymbol{x}')$ vs $H_A : p(\boldsymbol{x}) \neq q(\boldsymbol{x}')$. Chiefly, we explore the following design considerations: (i) what **representation** to run the test on; (ii) which **two-sample test** to run; (iii) when the representation is multidimensional; whether to run **multivariate or multiple univariate two-sample tests**; and (iv) **how to combine** their results.

## 3.1   Dimensionality Reduction

We now introduce the multiple dimensionality reduction (DR) techniques that we compare vis-a-vis their effectiveness in shift detection (in concert with two-sample testing). Note that absent assumptions on the data, these mappings, which reduce the data dimensionality from $D$ to $K$ (with $K \ll D$), are in general surjective, with many inputs mapping to the same output. Thus, it is trivial to construct pathological cases where the distribution of inputs shifts while the distribution of low-dimensional latent representations remains fixed, yielding false negatives. However, we speculate that in a non-adversarial setting, such shifts may be exceedingly unlikely. Thus our approach is (i) empirically motivated; and (ii) not put forth as a defense against worst-case adversarial attacks.

**No Reduction (*NoRed* ◯):** To justify the use of any DR technique, our default baseline is to run tests on the original raw features.

**Principal Components Analysis (*PCA* ◌):** Principal components analysis is a standard tool that finds an optimal orthogonal transformation matrix $\boldsymbol{R}$ such that points are linearly uncorrelated after transformation. This transformation is learned in such a way that the first principal component accounts for as much of the variability in the dataset as possible, and that each succeeding principal component captures as much of the remaining variance as possible subject to the constraint that it be orthogonal to the preceding components. Formally, we wish to learn $\boldsymbol{R}$ given $\boldsymbol{X}$ under the mentioned constraints such that $\hat{\boldsymbol{X}} = \boldsymbol{X}\boldsymbol{R}$ yields a more compact data representation.

**Sparse Random Projection (*SRP* ◠):** Since computing the optimal transformation might be expensive in high dimensions, random projections are a popular DR technique which trade a controlled amount of accuracy for faster processing times. Specifically, we make use of sparse random projections, a more memory- and computationally-efficient modification of standard Gaussian random projections. Formally, we generate a random projection matrix $\boldsymbol{R}$ and use it to reduce the dimensionality of a given data matrix $\boldsymbol{X}$, such that $\hat{\boldsymbol{X}} = \boldsymbol{X}\boldsymbol{R}$. The elements of $\boldsymbol{R}$ are generated using the following rule set [1, 27]:

$$R_{ij} = \begin{cases} +\sqrt{\frac{v}{K}} & \text{with probability } \frac{1}{2v} \\ 0 & \text{with probability } 1 - \frac{1}{v} \\ -\sqrt{\frac{v}{K}} & \text{with probability } \frac{1}{2v} \end{cases} \qquad \text{where} \qquad v = \frac{1}{\sqrt{D}}. \qquad (1)$$

**Autoencoders (*TAE* ◇ and *UAE* ☐):** We compare the above-mentioned linear models to non-linear reduced-dimension representations using both *trained* (TAE) and *untrained* autoencoders (UAE). Formally, an autoencoder consists of an encoder function $\phi : \mathcal{X} \to \mathcal{H}$ and a decoder function $\psi : \mathcal{H} \to \mathcal{X}$ where the latent space $\mathcal{H}$ has lower dimensionality than the input space $\mathcal{X}$. As part of the training process, both the encoding function $\phi$ and the decoding function $\psi$ are learned jointly to reduce the reconstruction loss: $\phi, \psi = \arg\min_{\phi,\psi} \|\boldsymbol{X} - (\psi \circ \phi)\boldsymbol{X}\|^2$.

**Label Classifiers (*BBSDs* ◁ and *BBSDh* ▷):** Motivated by recent results achieved by black box shift detection (BBSD) [29], we also propose to use the outputs of a (deep network) *label classifier* trained on source data as our dimensionality-reduced representation. We explore variants using either the softmax outputs (BBSDs) or the hard-thresholded predictions (BBSDh) for subsequent two-sample testing. Since both variants provide differently sized output (with BBSDs providing an entire softmax vector and BBSDh providing a one-dimensional class prediction), different statistical tests are carried out on these representations.

**Domain Classifier (*Classif* ✕):** Here, we attempt to detect shift by explicitly training a *domain classifier* to discriminate between data from source and target domains. To this end, we partition both the source data and target data into two halves, using the first to train a domain classifier to distinguish source (class 0) from target (class 1) data. We then apply this model to the second

half and subsequently conduct a significance test to determine if the classifier's performance is statistically different from random chance.

## 3.2 Statistical Hypothesis Testing

The DR techniques each yield a representation, either uni- or multi-dimensional, and either continuous or discrete, depending on the method. The next step is to choose a suitable statistical hypothesis test for each of these representations.

**Multivariate Kernel Two-Sample Tests: Maximum Mean Discrepancy (MMD)**: For all multi-dimensional representations, we evaluate the Maximum Mean Discrepancy [16], a popular kernel-based technique for multivariate two-sample testing. MMD allows us to distinguish between two probability distributions $p$ and $q$ based on the mean embeddings $\boldsymbol{\mu}_p$ and $\boldsymbol{\mu}_q$ of the distributions in a reproducing kernel Hilbert space $\mathcal{F}$, formally

$$\text{MMD}(\mathcal{F}, p, q) = ||\boldsymbol{\mu}_p - \boldsymbol{\mu}_q||_{\mathcal{F}}^2. \tag{2}$$

Given samples from both distributions, we can calculate an unbiased estimate of the squared MMD statistic as follows

$$\text{MMD}^2 = \frac{1}{m^2 - m} \sum_{i=1}^{m} \sum_{j \neq i}^{m} \kappa(\boldsymbol{x}_i, \boldsymbol{x}_j) + \frac{1}{n^2 - n} \sum_{i=1}^{n} \sum_{j \neq i}^{n} \kappa(\boldsymbol{x}_i', \boldsymbol{x}_j') - \frac{2}{mn} \sum_{i=1}^{m} \sum_{j=1}^{n} \kappa(\boldsymbol{x}_i, \boldsymbol{x}_j') \tag{3}$$

where we use a squared exponential kernel $\kappa(\boldsymbol{x}, \tilde{\boldsymbol{x}}) = e^{-\frac{1}{\sigma}||\boldsymbol{x} - \tilde{\boldsymbol{x}}||^2}$ and set $\sigma$ to the median distance between points in the aggregate sample over $p$ and $q$ [16]. A $p$-value can then be obtained by carrying out a permutation test on the resulting kernel matrix.

**Multiple Univariate Testing: Kolmogorov-Smirnov (KS) Test + Bonferroni Correction**: As a simple baseline alternative to MMD, we consider the approach consisting of testing each of the $K$ dimensions separately (instead testing over all dimensions jointly). Here, for continuous data, we adopt the Kolmogorov-Smirnov (KS) test, a non-parametric test whose statistic is calculated by computing the largest difference $Z$ of the cumulative density functions (CDFs) over all values $\boldsymbol{z}$ as follows

$$Z = \sup_{\boldsymbol{z}} |F_p(\boldsymbol{z}) - F_q(\boldsymbol{z})| \tag{4}$$

where $F_p$ and $F_q$ are the empirical CDFs of the source and target data, respectively. Under the null hypothesis, $Z$ follows the Kolmogorov distribution.

Since we carry out a KS test on each of the $K$ components, we must subsequently combine the $p$-values from each test, raising the issue of multiple hypothesis testing. As we cannot make strong assumptions about the (in)dependence among the tests, we rely on a conservative aggregation method, notably the Bonferroni correction [4], which rejects the null hypothesis if the minimum $p$-value among all tests is less than $\alpha/K$ (where $\alpha$ is the significance level of the test). While several less conservative aggregations methods have been proposed [18, 32, 46, 53, 55], they typically require assumptions on the dependencies among the tests.

**Categorical Testing: Chi-Squared Test**: For the hard-thresholded label classifier (BBSDh), we employ Pearson's chi-squared test, a parametric tests designed to evaluate whether the frequency distribution of certain events observed in a sample is consistent with a particular theoretical distribution. Specifically, we use a test of homogeneity between the class distributions (expressed in a contingency table) of source and target data. The testing problem can be formalized as follows: Given a contingency table with 2 rows (one for absolute source and one for absolute target class frequencies) and $C$ columns (one for each of the $C$-many classes) containing observed counts $O_{ij}$, the expected frequency under the independence hypothesis for a particular cell is $E_{ij} = N_{\text{sum}} p_{i\bullet} p_{\bullet j}$ with $N_{\text{sum}}$ being the sum of all cells in the table, $p_{i\bullet} = \frac{O_{i\bullet}}{N_{\text{sum}}} = \sum_{j=1}^{C} \frac{O_{ij}}{N_{\text{sum}}}$ being the fraction of row totals, and $p_{\bullet j} = \frac{O_{\bullet j}}{N_{\text{sum}}} = \sum_{i=1}^{2} \frac{O_{ij}}{N_{\text{sum}}}$ being the fraction of column totals. The relevant test statistic $X^2$ can be computed as

$$X^2 = \sum_{i=1}^{2} \sum_{j=1}^{C} \frac{(O_{ij} - E_{ij})^2}{E_{ij}} \tag{5}$$

which, under the null hypothesis, follows a chi-squared distribution with $C - 1$ degrees of freedom: $X^2 \sim \chi_{C-1}^2$.

**Binomial Testing**: For the domain classifier, we simply compare its accuracy (acc) on held-out data to random chance via a binomial test. Formally, we set up a testing problem $H_0 : \text{acc} = 0.5$ vs $H_A : \text{acc} \neq 0.5$. Under the null hypothesis, the accuracy of the classifier follows a binomial distribution: $\text{acc} \sim \text{Bin}(N_{\text{hold}}, 0.5)$, where $N_{\text{hold}}$ corresponds to the number of held-out samples.

### 3.3  Obtaining Most Anomalous Samples

As our detection framework does not detect outliers but rather aims at capturing top-level shift dynamics, it is not possible for us to decide whether any given sample is in- or out-of-distribution. However, we can still provide an indication of what typical samples from the shifted distribution look like by harnessing domain assignments from the domain classifier. Specifically, we can identify the exemplars which the classifier was most confident in assigning to the target domain. Since the domain classifier assigns class-assignment confidence scores to each incoming sample via the softmax-layer at its output, it is easy to create a ranking of samples that are most confidently believed to come from the target domain (or, alternatively, from the source domain). Hence, whenever the binomial test signals a statistically significant accuracy deviation from chance, we can use use the domain classifier to obtain the most anomalous samples and present them to the user.

In contrast to the domain classifier, the other shift detectors do not base their shift detection potential on explicitly deciding which domain a single sample belongs to, instead comparing entire distributions against each other. While we did explore initial ideas on identifying samples which if removed would lead to a large increase in the overall $p$-value, the results we obtained were unremarkable.

### 3.4  Determining the Malignancy of a Shift

Theoretically, absent further assumptions, distribution shifts can cause arbitrarily severe degradation in performance. However, in practice distributions shift constantly, and often these changes are benign. Practitioners should therefore be interested in distinguishing malignant shifts that damage predictive performance from benign shifts that negligibly impact performance. Although prediction quality can be assessed easily on source data on which the black-box model $f$ was trained, we are not able compute the target error directly without labels.

We therefore explore a heuristic method for approximating the target performance by making use of the domain classifier's class assignments as follows: Given access to a labeling function that can correctly label samples, we can feed in those examples predicted by the domain classifier as likely to come from the target domain. We can then compare these (true) labels to the labels returned by the black box model $f$ by feeding it the same anomalous samples. If our model is inaccurate on these examples (where the exact threshold can be user-specified to account for varying sensitivities to accuracy drops), then we ought to be concerned that the shift is malignant. Put simply, we suggest evaluating the accuracy of our models on precisely those examples which are most confidently assigned to the target domain.

## 4  Experiments

Our main experiments were carried out on the MNIST ($N_{\text{tr}} = 50000$; $N_{\text{val}} = 10000$; $N_{\text{te}} = 10000$; $D = 28 \times 28 \times 1$; $C = 10$ classes) [25] and CIFAR-10 ($N_{\text{tr}} = 40000$; $N_{\text{val}} = 10000$; $N_{\text{te}} = 10000$; $D = 32 \times 32 \times 3$; $C = 10$ classes) [23] image datasets. For the autoencoder (UAE & TAE) experiments, we employ a convolutional architecture with 3 convolutional layers and 1 fully-connected layer. For both the label and the domain classifier we use a ResNet-18 [17]. We train all networks (TAE, BBSDs, BBSDh, Classif) using stochastic gradient descent with momentum in batches of 128 examples over 200 epochs with early stopping.

For PCA, SRP, UAE, and TAE, we reduce dimensionality to $K = 32$ latent dimensions, which for PCA explains roughly $80\%$ of the variance in the CIFAR-10 dataset. The label classifier BBSDs reduces dimensionality to the number of classes $C$. Both the hard label classifier BBSDh and the domain classifier Classif reduce dimensionality to a one-dimensional class prediction, where BBSDh predicts label assignments and Classif predicts domain assignments.

To challenge our detection methods, we simulate a variety of shifts, affecting both the covariates and the label proportions. For all shifts, we evaluate the various methods' abilities to detect shift at

a significance level of $\alpha = 0.05$. We also include the no-shift case to check against false positives. We randomly split all of the data into training, validation, and test sets according to the indicated proportions $N_{tr}$, $N_{val}$, and $N_{te}$ and then apply a particular shift to the test set only. In order to qualitatively quantify the robustness of our findings, shift detection performance is averaged over a total of 5 random splits, which ensures that we apply the same type of shift to different subsets of the data. The selected training data used to fit the DR methods is kept constant across experiments with only the splits between validation and test changing across the random runs. Note that DR methods are learned using training data, while shift detection is being performed on dimensionality-reduced representations of the validation and the test set. We evaluate the models with various amounts of samples from the test set $s \in \{10, 20, 50, 100, 200, 500, 1000, 10000\}$. Because of the unfavorable dependence of kernel methods on the dataset size, we run these methods only up until 1000 target samples have been acquired.

For each shift type (as appropriate) we explored three levels of shift intensity (e.g. the magnitude of added noise) and various percentages of affected data $\delta \in \{0.1, 0.5, 1.0\}$. Specifically, we explore the following types of shifts:

(a) **Adversarial (*adv*)**: We turn a fraction $\delta$ of samples into adversarial samples via FGSM [13];

(b) **Knock-out (*ko*)**: We remove a fraction $\delta$ of samples from class 0, creating class imbalance [29];

(c) **Gaussian noise (*gn*)**: We corrupt covariates of a fraction $\delta$ of test set samples by Gaussian noise with standard deviation $\sigma \in \{1, 10, 100\}$ (denoted *s_gn*, *m_gn*, and *l_gn*);

(d) **Image (*img*)**: We also explore more natural shifts to images, modifying a fraction $\delta$ of images with combinations of random rotations $\{10, 40, 90\}$, $(x, y)$-axis-translation percentages $\{0.05, 0.2, 0.4\}$, as well as zoom-in percentages $\{0.1, 0.2, 0.4\}$ (denoted *s_img*, *m_img*, and *l_img*);

(e) **Image + knock-out (*m_img+ko*)**: We apply a fixed medium image shift with $\delta_1 = 0.5$ and a variable knock-out shift $\delta$;

(f) **Only-zero + image (*oz+m_img*)**: Here, we only include images from class 0 in combination with a variable medium image shift affecting only a fraction $\delta$ of the data;

(g) **Original splits**: We evaluate our detectors on the original source/target splits provided by the creators of MNIST, CIFAR-10, Fashion MNIST [54], and SVHN [35] datasets (assumed to be i.i.d.);

(h) **Domain adaptation datasets**: Data from the domain adaptation task transferring from MNIST (source) to USPS (target) ($N_{tr} = N_{val} = N_{te} = 1000$; $D = 16 \times 16 \times 1$; $C = 10$ classes) [31] as well as the COIL-100 dataset ($N_{tr} = N_{val} = N_{te} = 2400$; $D = 32 \times 32 \times 3$; $C = 100$ classes) [34] where images between $0°$ and $175°$ are sampled by the source and images between $180°$ and $355°$ are sampled by the target distribution.

We provide a sample implementation of our experiments-pipeline written in Python, making use of sklearn [36] and Keras [11], located at: `https://github.com/steverab/failing-loudly`.

## 5 Discussion

**Univariate VS Multivariate Tests**: We first evaluate whether we can detect shifts more easily using multiple univariate tests and aggregating their results via the Bonferroni correction or by using multivariate kernel tests. We were surprised to find that, despite the heavy correction, multiple univariate testing seem to offer comparable performance to multivariate testing (see Table 1a).

**Dimensionality Reduction Methods**: For each testing method and experimental setting, we evaluate which DR technique is best suited to shift detection. Specifically in the multiple-univariate-testing case (and overall), BBSDs was the best-performing DR method. In the multivariate-testing case, UAE performed best. In both cases, these methods consistently outperformed others across sample sizes. The domain classifier, a popular shift detection approach, performs badly in the low-sample regime ($\leq 100$ samples), but catches up as more samples are obtained. Noticeably, the multivariate test performs poorly in the no reduction case, which is also regarded a widely used shift detection baseline. Table 1a summarizes these results.

We note that BBSDs being the best overall method for detecting shift is good news for ML practitioners. When building black-box models with the main purpose of classification, said model can be

Table 1: Dimensionality reduction methods (a) and shift-type (b) comparison. Underlined entries indicate accuracy values larger than 0.5.

(a) Detection accuracy of different dimensionality reduction techniques across all simulated shifts on MNIST and CIFAR-10. **Green bold** entries indicate the best DR method at a given sample size, *red italic* the worst. Results for $\chi^2$ and Bin tests are only reported once under the univariate category. BBSDs performs best for univariate testing, while both UAE and TAE perform best for multivariate testing.

| Test | DR | Number of samples from test | | | | | | | |
|---|---|---|---|---|---|---|---|---|---|
| | | 10 | 20 | 50 | 100 | 200 | 500 | 1,000 | 10,000 |
| Univ. tests | NoRed | 0.03 | 0.15 | 0.26 | 0.36 | 0.41 | 0.47 | 0.54 | 0.72 |
| | *PCA* | 0.11 | 0.15 | 0.30 | 0.36 | 0.41 | 0.46 | 0.54 | 0.63 |
| | SRP | 0.15 | 0.15 | 0.23 | 0.27 | 0.34 | 0.42 | 0.55 | 0.68 |
| | UAE | 0.12 | 0.16 | 0.27 | 0.33 | 0.41 | 0.49 | 0.56 | 0.77 |
| | TAE | 0.18 | 0.23 | 0.31 | 0.38 | 0.43 | 0.47 | 0.55 | 0.69 |
| | **BBSDs** | **0.19** | **0.28** | **0.47** | **0.47** | **0.51** | **0.65** | **0.70** | **0.79** |
| $\chi^2$ | *BBSDh* | 0.03 | 0.07 | 0.12 | 0.22 | *0.22* | *0.40* | *0.46* | *0.57* |
| Bin | *Classif* | *0.01* | *0.03* | *0.11* | *0.21* | 0.28 | 0.42 | 0.51 | 0.67 |
| Multiv. tests | NoRed | 0.14 | *0.15* | *0.22* | *0.28* | 0.32 | *0.44* | 0.55 | – |
| | PCA | 0.15 | 0.18 | 0.33 | 0.38 | 0.40 | 0.46 | 0.55 | – |
| | SRP | *0.12* | 0.18 | 0.23 | 0.31 | *0.31* | *0.44* | 0.54 | – |
| | **UAE** | **0.20** | **0.27** | **0.40** | **0.43** | **0.45** | **0.53** | **0.61** | – |
| | TAE | 0.18 | 0.26 | 0.37 | 0.38 | 0.45 | 0.52 | 0.59 | – |
| | BBSDs | 0.16 | 0.20 | 0.25 | 0.35 | 0.35 | 0.47 | *0.50* | – |

(b) Detection accuracy of different shifts on MNIST and CIFAR-10 using the best-performing DR technique (univariate: BBSDs, multivariate: UAE). **Green bold** shifts are identified as harmless, *red italic* shifts as harmful.

| Test | Shift | Number of samples from test | | | | | | | |
|---|---|---|---|---|---|---|---|---|---|
| | | 10 | 20 | 50 | 100 | 200 | 500 | 1,000 | 10,000 |
| Univariate BBSDs | **s_gn** | 0.00 | 0.00 | 0.03 | 0.03 | 0.07 | 0.10 | 0.10 | 0.10 |
| | **m_gn** | 0.00 | 0.00 | 0.10 | 0.13 | 0.13 | 0.13 | 0.23 | 0.37 |
| | **l_gn** | 0.17 | 0.27 | 0.53 | 0.63 | 0.67 | 0.83 | 0.87 | 1.00 |
| | **s_img** | 0.00 | 0.00 | 0.23 | 0.30 | 0.40 | 0.63 | 0.70 | 0.93 |
| | *m_img* | 0.30 | 0.37 | 0.60 | 0.67 | 0.70 | 0.80 | 0.90 | 1.00 |
| | *l_img* | 0.30 | 0.50 | 0.70 | 0.70 | 0.77 | 0.87 | 0.97 | 1.00 |
| | *adv* | 0.13 | 0.27 | 0.40 | 0.43 | 0.53 | 0.77 | 0.83 | 0.90 |
| | **ko** | 0.00 | 0.00 | 0.07 | 0.07 | 0.07 | 0.33 | 0.40 | 0.70 |
| | *m_img+ko* | 0.13 | 0.40 | 0.87 | 0.93 | 0.90 | 1.00 | 1.00 | 1.00 |
| | *oz+m_img* | 0.67 | 1.00 | 1.00 | 1.00 | 1.00 | 1.00 | 1.00 | 1.00 |
| Multivariate UAE | s_gn | 0.03 | 0.03 | 0.03 | 0.03 | 0.03 | 0.07 | 0.07 | – |
| | m_gn | 0.03 | 0.03 | 0.03 | 0.03 | 0.17 | 0.27 | 0.30 | – |
| | l_gn | 0.50 | 0.57 | 0.67 | 0.70 | 0.80 | 0.90 | 1.00 | – |
| | s_img | 0.17 | 0.20 | 0.27 | 0.30 | 0.40 | 0.47 | 0.63 | – |
| | m_img | 0.23 | 0.33 | 0.37 | 0.40 | 0.47 | 0.60 | 0.70 | – |
| | l_img | 0.30 | 0.30 | 0.37 | 0.47 | 0.60 | 0.77 | 0.87 | – |
| | adv | 0.03 | 0.20 | 0.27 | 0.27 | 0.33 | 0.40 | 0.40 | – |
| | ko | 0.10 | 0.13 | 0.13 | 0.13 | 0.17 | 0.17 | 0.30 | – |
| | m_img+ko | 0.20 | 0.30 | 0.37 | 0.53 | 0.54 | 0.63 | 0.87 | – |
| | oz+m_img | 0.27 | 0.63 | 0.77 | 1.00 | 1.00 | 1.00 | 1.00 | – |

Table 2: Shift detection performance based on shift intensity (a) and perturbed sample percentages (b) using the best-performing DR technique (univariate: BBSDs, multivariate: UAE). Underlined entries indicate accuracy values larger than 0.5.

(a) Detection accuracy of varying shift intensities.

| Test | Intensity | Number of samples from test | | | | | | | |
|---|---|---|---|---|---|---|---|---|---|
| | | 10 | 20 | 50 | 100 | 200 | 500 | 1,000 | 10,000 |
| Univ. | Small | 0.00 | 0.00 | 0.14 | 0.14 | 0.18 | 0.36 | 0.40 | 0.54 |
| | Medium | 0.14 | 0.21 | 0.39 | 0.38 | 0.42 | 0.57 | 0.66 | 0.76 |
| | Large | 0.32 | 0.54 | 0.78 | 0.82 | 0.83 | 0.92 | 0.96 | 1.00 |
| Multiv. | Small | 0.11 | 0.11 | 0.12 | 0.14 | 0.20 | 0.23 | 0.33 | – |
| | Medium | 0.11 | 0.19 | 0.23 | 0.27 | 0.32 | 0.42 | 0.44 | – |
| | Large | 0.34 | 0.45 | 0.57 | 0.68 | 0.72 | 0.82 | 0.93 | – |

(b) Detection accuracy of varying shift percentages.

| Test | Percentage | Number of samples from test | | | | | | | |
|---|---|---|---|---|---|---|---|---|---|
| | | 10 | 20 | 50 | 100 | 200 | 500 | 1,000 | 10,000 |
| Univ. | 10% | 0.11 | 0.15 | 0.24 | 0.25 | 0.28 | 0.44 | 0.54 | 0.66 |
| | 50% | 0.14 | 0.28 | 0.52 | 0.53 | 0.60 | 0.68 | 0.72 | 0.85 |
| | 100% | 0.26 | 0.41 | 0.61 | 0.64 | 0.70 | 0.82 | 0.84 | 0.86 |
| Multiv. | 10% | 0.12 | 0.13 | 0.21 | 0.26 | 0.27 | 0.31 | 0.44 | – |
| | 50% | 0.19 | 0.27 | 0.41 | 0.41 | 0.47 | 0.57 | 0.60 | – |
| | 100% | 0.29 | 0.41 | 0.44 | 0.53 | 0.60 | 0.70 | 0.78 | – |

easily extended to also double as a shift detector. Moreover, black-box models with soft predictions that were built and trained in the past can be turned into shift detectors retrospectively.

**Shift Types**: Table 1b lists shift detection accuracy values for each distinct shift as an increasing amount of samples is obtained from the target domain. Specifically, we see that *l_gn*, *m_gn*, *l_img*, *m_img+ko*, *oz+m_img*, and even *adv* are easily detectable, many of them even with few samples, while *s_gn*, *m_gn*, and *ko* are hard to detect even with many samples. With a few exceptions, the best DR technique (BBDSs for multiple univariate tests, UAE for multivariate tests) is significantly faster and more accurate at detecting shift than the average of all dimensionality reduction methods.

**Shift Strength**: Based on the results in Table 2a, we can conclude that small shifts (*s_gn*, *s_img*, and *ko*) are harder to detect than medium shifts (*m_gn*, *m_img*, and *adv*) which in turn are harder to detect than large shifts (*l_gn*, *l_img*, *m_img+ko*, and *oz+m_img*). Specifically, we see that large shifts can on average already be detected with better than chance accuracy at only 20 samples using BBSDs, while medium and small shifts require orders of magnitude more samples in order to achieve similar accuracy. Moreover, the results in Table 2b show that while target data exhibiting only 10% anomalous samples are hard to detect, suggesting that this setting might be better addressed via outlier detection, perturbation percentages 50% and 100% can already be detected with better than chance accuracy using 50 samples.

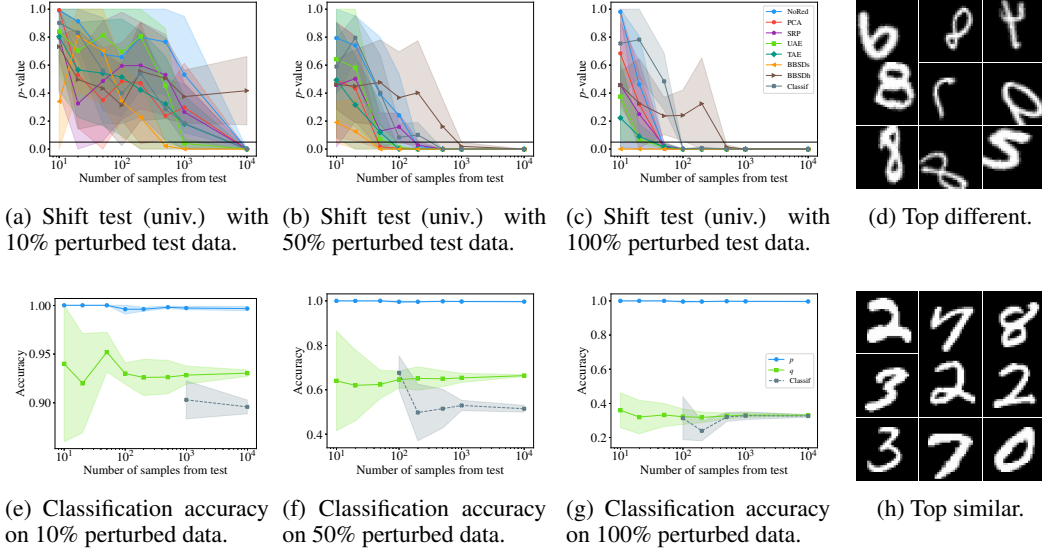

(a) Shift test (univ.) with 10% perturbed test data.

(b) Shift test (univ.) with 50% perturbed test data.

(c) Shift test (univ.) with 100% perturbed test data.

(d) Top different.

(e) Classification accuracy on 10% perturbed data.

(f) Classification accuracy on 50% perturbed data.

(g) Classification accuracy on 100% perturbed data.

(h) Top similar.

Figure 2: Shift detection results for medium image shift on MNIST. Subfigures (a)-(c) show the $p$-value evolution of the different DR methods with varying percentages of perturbed data, while subfigures (e)-(g) show the obtainable accuracies over the same perturbations. Subfigures (d) and (h) show the *most different* and *most similar* exemplars returned by the domain classifier across perturbation percentages. Plots show mean values obtained over 5 random runs with a 1-$\sigma$ error-bar.

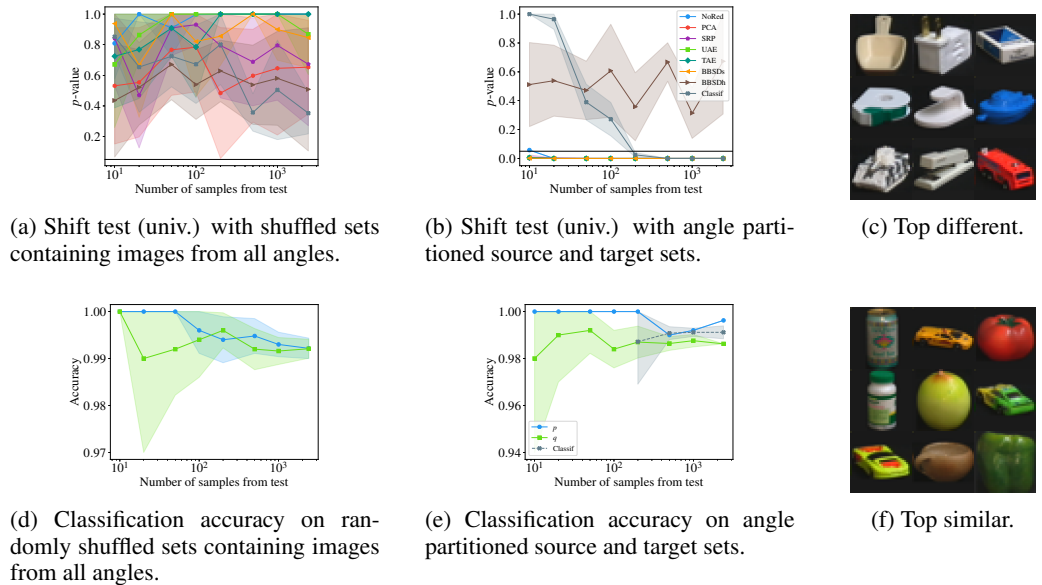

(a) Shift test (univ.) with shuffled sets containing images from all angles.

(b) Shift test (univ.) with angle partitioned source and target sets.

(c) Top different.

(d) Classification accuracy on randomly shuffled sets containing images from all angles.

(e) Classification accuracy on angle partitioned source and target sets.

(f) Top similar.

Figure 3: Shift detection results on COIL-100 dataset. Subfigure organization is similar to Figure 2.

**Most Anomalous Samples and Shift Malignancy**: Across all experiments, we observe that the most different and most similar examples returned by the domain classifier are useful in characterizing the shift. Furthermore, we can successfully distinguish malignant from benign shifts (as reported in Table 1b) by using the framework proposed in Section 3.4. While we recognize that having access to an external labeling function is a strong assumption and that accessing all true labels would be prohibitive at deployment, our experimental results also showed that, compared to the total sample size, two to three orders of magnitude fewer labeled examples suffice to obtain a good approximation of the (usually unknown) target accuracy.

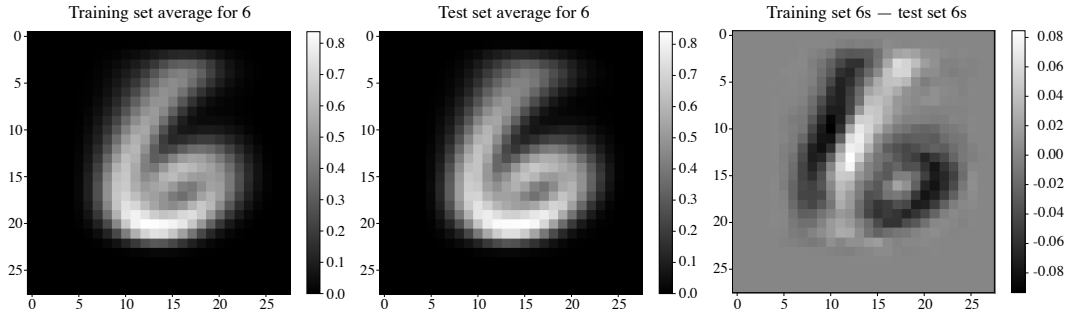

Figure 4: Difference plot for training and test set sixes.

**Individual Examples**: While full results with exact $p$-value evolution and anomalous samples are documented in the supplementary material, we briefly present two illustrative results in detail:

(a) *Synthetic medium image shift on MNIST (Figure 2)*: From subfigures (a)-(c), we see that most methods are able to detect the simulated shift with BBSDs being the quickest method for all tested perturbation percentages. We further observe in subfigures (e)-(g) that the (true) accuracy on samples from $q$ increasingly deviates from the model's performance on source data from $p$ as more samples are perturbed. Since true target accuracy is usually unknown, we use the accuracy obtained on the top anomalous labeled instances returned by the domain classifier Classif. As we can see, these values significantly deviate from accuracies obtained on $p$, which is why we consider this shift harmful to the label classifier's performance.

(b) *Rotation angle partitioning on COIL-100 (Figure 3)*: Subfigures (a) and (b) show that our testing framework correctly claims the randomly shuffled dataset containing images from all angles to not contain a shift, while it identifies the partitioned dataset to be noticeably different. However, as we can see from subfigure (e), this shift does not harm the classifier's performance, meaning that the classifier can safely be deployed even when encountering this specific dataset shift.

**Original Splits**: According to our tests, the original split from the MNIST dataset appears to exhibit a dataset shift. After inspecting the most anomalous samples returned by the domain classifier, we observed that many of these samples depicted the digit 6. A mean-difference plot (see Figure 4) between sixes from the training set and sixes from the test set revealed that the training instances are rotated slightly to the right, while the test samples are drawn more open and centered. To back up this claim even further, we also carried out a two-sample KS test between the two sets of sixes in the input space and found that the two sets can conclusively be regarded as different with a $p$-value of $2.7 \cdot 10^{-10}$, significantly undercutting the respective Bonferroni threshold of $6.3 \cdot 10^{-5}$. While this specific shift does not look particularly significant to the human eye (and is also declared harmless by our malignancy detector), this result however still shows that the original MNIST split is not i.i.d.

## 6   Conclusions

In this paper, we put forth a comprehensive empirical investigation, examining the ways in which dimensionality reduction and two-sample testing might be combined to produce a practical pipeline for detecting distribution shift in real-life machine learning systems. Our results yielded the surprising insights that (i) black-box shift detection with soft predictions works well across a wide variety of shifts, even when some of its underlying assumptions do not hold; (ii) that aggregated univariate tests performed separately on each latent dimension offer comparable shift detection performance to multivariate two-sample tests; and (iii) that harnessing predictions from domain-discriminating classifiers enables characterization of a shift's type and its malignancy. Moreover, we produced the surprising observation that the MNIST dataset, despite ostensibly representing a random split, exhibits a significant (although not worrisome) distribution shift.

Our work suggests several open questions that might offer promising paths for future work, including (i) shift detection for online data, which would require us to account for and exploit the high degree of correlation between adjacent time steps [22]; and, since we have mostly explored a standard image classification setting for our experiments, (ii) applying our framework to other machine learning domains such as natural language processing or graphs.

**Acknowledgements**

We thank the Center for Machine Learning and Health, a joint venture of Carnegie Mellon University, UPMC, and the University of Pittsburgh for supporting our collaboration with Abridge AI to develop robust models for machine learning in healthcare. We are also grateful to Salesforce Research, Facebook AI Research, and Amazon AI for their support of our work on robust deep learning under distribution shift.

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
