[Supplementary Material]

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

# A Detailed Shift Detection Results

Our complete shift detection results in which we evaluate different kinds of target shifts on MNIST and CIFAR-10 using the proposed methods are documented below. In addition to our artificially generated shifts, we also evaluated our testing procedure on the original splits provided by MNIST, Fashion MNIST, CIFAR-10, and SVHN.

## A.1 Artificially Generated Shifts

### A.1.1 MNIST

(a) 10% adversarial samples.  (b) 50% adversarial samples.  (c) 100% adversarial samples.

(d) 10% adversarial samples.  (e) 50% adversarial samples.  (f) 100% adversarial samples.

(g) Top different samples.  (h) Top similar samples.

Figure 5: MNIST adversarial shift, univariate two-sample tests + Bonferroni aggregation.

(a) 10% adversarial samples.  (b) 50% adversarial samples.  (c) 100% adversarial samples.

Figure 6: MNIST adversarial shift, multivariate two-sample tests.

(a) Knock out 10% of class 0.
(b) Knock out 50% of class 0.
(c) Knock out 100% of class 0.

(d) Knock out 10% of class 0.
(e) Knock out 50% of class 0.
(f) Knock out 100% of class 0.

(g) Top different samples.
(h) Top similar samples.

Figure 7: MNIST knock-out shift, univariate two-sample tests + Bonferroni aggregation.

(a) Knock out 10% of class 0.
(b) Knock out 50% of class 0.
(c) Knock out 100% of class 0.

Figure 8: MNIST knock-out shift, multivariate two-sample tests.

(a) 10% perturbed samples.

(b) 50% perturbed samples.

(c) 100% perturbed samples.

(d) 10% perturbed samples.

(e) 50% perturbed samples.

(f) 100% perturbed samples.

(g) Top different samples.

(h) Top similar samples.

Figure 9: MNIST large Gaussian noise shift, univariate two-sample tests + Bonferroni aggregation.

(a) 10% perturbed samples.

(b) 50% perturbed samples.

(c) 100% perturbed samples.

Figure 10: MNIST large Gaussian noise shift, multivariate two-sample tests.

(a) 10% perturbed samples.

(b) 50% perturbed samples.

(c) 100% perturbed samples.

(d) 10% perturbed samples.

(e) 50% perturbed samples.

(f) 100% perturbed samples.

(g) Top different samples.

(h) Top similar samples.

Figure 11: MNIST medium Gaussian noise shift, univariate two-sample tests + Bonferroni aggregation.

(a) 10% perturbed samples.

(b) 50% perturbed samples.

(c) 100% perturbed samples.

Figure 12: MNIST medium Gaussian noise shift, multivariate two-sample tests.

(a) 10% perturbed samples.

(b) 50% perturbed samples.

(c) 100% perturbed samples.

(d) 10% perturbed samples.

(e) 50% perturbed samples.

(f) 100% perturbed samples.

(g) Top different samples.

(h) Top similar samples.

Figure 13: MNIST small Gaussian noise shift, univariate two-sample tests + Bonferroni aggregation.

(a) 10% perturbed samples.

(b) 50% perturbed samples.

(c) 100% perturbed samples.

Figure 14: MNIST small Gaussian noise shift, multivariate two-sample tests.

(a) 10% perturbed samples.     (b) 50% perturbed samples.     (c) 100% perturbed samples.

(d) 10% perturbed samples.     (e) 50% perturbed samples.     (f) 100% perturbed samples.

(g) Top different samples.           (h) Top similar samples.

Figure 15: MNIST large image shift, univariate two-sample tests + Bonferroni aggregation.

(a) 10% perturbed samples.     (b) 50% perturbed samples.     (c) 100% perturbed samples.

Figure 16: MNIST large image shift, multivariate two-sample tests.

(a) 10% perturbed samples.

(b) 50% perturbed samples.

(c) 100% perturbed samples.

(d) 10% perturbed samples.

(e) 50% perturbed samples.

(f) 100% perturbed samples.

(g) Top different samples.

(h) Top similar samples.

Figure 17: MNIST medium image shift, univariate two-sample tests + Bonferroni aggregation.

(a) 10% perturbed samples.

(b) 50% perturbed samples.

(c) 100% perturbed samples.

Figure 18: MNIST medium image shift, multivariate two-sample tests.

(a) 10% perturbed samples.　(b) 50% perturbed samples.　(c) 100% perturbed samples.

(d) 10% perturbed samples.　(e) 50% perturbed samples.　(f) 100% perturbed samples.

(g) Top different samples.　(h) Top similar samples.

Figure 19: MNIST small image shift, univariate two-sample tests + Bonferroni aggregation.

(a) 10% perturbed samples.　(b) 50% perturbed samples.　(c) 100% perturbed samples.

Figure 20: MNIST small image shift, multivariate two-sample tests.

(a) Knock out 10% of class 0.  (b) Knock out 50% of class 0.  (c) Knock out 100% of class 0.

(d) Knock out 10% of class 0.  (e) Knock out 50% of class 0.  (f) Knock out 100% of class 0.

(g) Top different samples.

(h) Top similar samples.

Figure 21: MNIST medium image shift (50%, fixed) plus knock-out shift (variable), univariate two-sample tests + Bonferroni aggregation.

(a) Knock out 10% of class 0.  (b) Knock out 50% of class 0.  (c) Knock out 100% of class 0.

Figure 22: MNIST medium image shift (50%, fixed) plus knock-out shift (variable), multivariate two-sample tests.

(a) 10% perturbed samples.

(b) 50% perturbed samples.

(c) 100% perturbed samples.

(d) 10% perturbed samples.

(e) 50% perturbed samples.

(f) 100% perturbed samples.

(g) Top different samples.

(h) Top similar samples.

Figure 23: MNIST only-zero shift (fixed) plus medium image shift (variable), univariate two-sample tests + Bonferroni aggregation.

(a) 10% perturbed samples.

(b) 50% perturbed samples.

(c) 100% perturbed samples.

Figure 24: MNIST only-zero shift (fixed) plus medium image shift (variable), multivariate two-sample tests.

(a) Randomly shuffled dataset with same split proportions as original dataset.

(b) Original split.

(c) Randomly shuffled dataset with same split proportions as original dataset.

(d) Original split.

(e) Top different samples.

(f) Top similar samples.

Figure 25: MNIST to USPS domain adaptation, univariate two-sample tests + Bonferroni aggregation.

(a) Randomly shuffled dataset with same split proportions as original dataset.

(b) Original split.

Figure 26: MNIST to USPS domain adaptation, multivariate two-sample tests.

(a) 10% adversarial samples.

(b) 50% adversarial samples.

(c) 100% adversarial samples.

(d) 10% adversarial samples.

(e) 50% adversarial samples.

(f) 100% adversarial samples.

(g) Top different samples.

(h) Top similar samples.

Figure 27: CIFAR-10 adversarial shift, univariate two-sample tests + Bonferroni aggregation.

(a) 10% adversarial samples.

(b) 50% adversarial samples.

(c) 100% adversarial samples.

Figure 28: CIFAR-10 adversarial shift, multivariate two-sample tests.

(a) Knock out 10% of class 0.

(b) Knock out 50% of class 0.

(c) Knock out 100% of class 0.

(d) Knock out 10% of class 0.

(e) Knock out 50% of class 0.

(f) Knock out 100% of class 0.

No samples available as *Classif* did not detect a shift.

No samples available as *Classif* did not detect a shift.

(g) Top different samples.

(h) Top similar samples.

Figure 29: CIFAR-10 knock-out shift, univariate two-sample tests + Bonferroni aggregation.

(a) Knock out 10% of class 0.

(b) Knock out 50% of class 0.

(c) Knock out 100% of class 0.

Figure 30: CIFAR-10 knock-out shift, multivariate two-sample tests.

(a) 10% perturbed samples.

(b) 50% perturbed samples.

(c) 100% perturbed samples.

(d) 10% perturbed samples.

(e) 50% perturbed samples.

(f) 100% perturbed samples.

(g) Top different samples.

(h) Top similar samples.

Figure 31: CIFAR-10 large Gaussian noise shift, univariate two-sample tests + Bonferroni aggregation.

(a) 10% perturbed samples.

(b) 50% perturbed samples.

(c) 100% perturbed samples.

Figure 32: CIFAR-10 large Gaussian noise shift, multivariate two-sample tests.

(a) 10% perturbed samples.

(b) 50% perturbed samples.

(c) 100% perturbed samples.

(d) 10% perturbed samples.

(e) 50% perturbed samples.

(f) 100% perturbed samples.

(g) Top different samples.

(h) Top similar samples.

Figure 33: CIFAR-10 medium Gaussian noise shift, univariate two-sample tests + Bonferroni aggregation.

(a) 10% perturbed samples.

(b) 50% perturbed samples.

(c) 100% perturbed samples.

Figure 34: CIFAR-10 medium Gaussian noise shift, multivariate two-sample tests.

(a) 10% perturbed samples.　　(b) 50% perturbed samples.　　(c) 100% perturbed samples.

(d) 10% perturbed samples.　　(e) 50% perturbed samples.　　(f) 100% perturbed samples.

No samples available as *Classif* did not detect a shift.

No samples available as *Classif* did not detect a shift.

(g) Top different samples.　　　　　　　　(h) Top similar samples.

Figure 35: CIFAR-10 small Gaussian noise shift, univariate two-sample tests + Bonferroni aggregation.

(a) 10% perturbed samples.　　(b) 50% perturbed samples.　　(c) 100% perturbed samples.

Figure 36: CIFAR-10 small Gaussian noise shift, multivariate two-sample tests.

(a) 10% perturbed samples.

(b) 50% perturbed samples.

(c) 100% perturbed samples.

(d) 10% perturbed samples.

(e) 50% perturbed samples.

(f) 100% perturbed samples.

(g) Top different samples.

(h) Top similar samples.

Figure 37: CIFAR-10 large image shift, univariate two-sample tests + Bonferroni aggregation.

(a) 10% perturbed samples.

(b) 50% perturbed samples.

(c) 100% perturbed samples.

Figure 38: CIFAR-10 large image shift, multivariate two-sample tests.

(a) 10% perturbed samples.

(b) 50% perturbed samples.

(c) 100% perturbed samples.

(d) 10% perturbed samples.

(e) 50% perturbed samples.

(f) 100% perturbed samples.

(g) Top different samples.

(h) Top similar samples.

Figure 39: CIFAR-10 medium image shift, univariate two-sample tests + Bonferroni aggregation.

(a) 10% perturbed samples.

(b) 50% perturbed samples.

(c) 100% perturbed samples.

Figure 40: CIFAR-10 medium image shift, multivariate two-sample tests.

(a) 10% perturbed samples.　　(b) 50% perturbed samples.　　(c) 100% perturbed samples.

(d) 10% perturbed samples.　　(e) 50% perturbed samples.　　(f) 100% perturbed samples.

(g) Top different samples.　　　　　　　　(h) Top similar samples.

Figure 41: CIFAR-10 small image shift, univariate two-sample tests + Bonferroni aggregation.

(a) 10% perturbed samples.　　(b) 50% perturbed samples.　　(c) 100% perturbed samples.

Figure 42: CIFAR-10 small image shift, multivariate two-sample tests.

(a) Knock out 10% of class 0.

(b) Knock out 50% of class 0.

(c) Knock out 100% of class 0.

(d) Knock out 10% of class 0.

(e) Knock out 50% of class 0.

(f) Knock out 100% of class 0.

(g) Top different samples.

(h) Top similar samples.

Figure 43: CIFAR-10 medium image shift (50%, fixed) plus knock-out shift (variable), univariate two-sample tests + Bonferroni aggregation.

(a) Knock out 10% of class 0.

(b) Knock out 50% of class 0.

(c) Knock out 100% of class 0.

Figure 44: CIFAR-10 medium image shift (50%, fixed) plus knock-out shift (variable), multivariate two-sample tests.

(a) 10% perturbed samples.

(b) 50% perturbed samples.

(c) 100% perturbed samples.

(d) 10% perturbed samples.

(e) 50% perturbed samples.

(f) 100% perturbed samples.

(g) Top different samples.

(h) Top similar samples.

Figure 45: CIFAR-10 only-zero shift (fixed) plus medium image shift (variable), univariate two-sample tests + Bonferroni aggregation.

(a) 10% perturbed samples.

(b) 50% perturbed samples.

(c) 100% perturbed samples.

Figure 46: CIFAR-10 only-zero shift (fixed) plus medium image shift (variable), multivariate two-sample tests.

## A.2 Original Splits

### A.2.1 MNIST

(a) Randomly shuffled dataset with same split proportions as original dataset.

(b) Original split.

(c) Randomly shuffled dataset with same split proportions as original dataset.

(d) Original split.

(e) Top different samples.

(f) Top similar samples.

Figure 47: MNIST randomized and original split, univariate two-sample tests + Bonferroni aggregation.

(a) Randomly shuffled dataset with same split proportions as original dataset.

(b) Original split.

Figure 48: MNIST randomized and original split, multivariate two-sample tests.

## A.2.2 Fashion MNIST

(a) Randomly shuffled dataset with same split proportions as original dataset.

(b) Original split.

(c) Randomly shuffled dataset with same split proportions as original dataset.

(d) Original split.

No samples available as *Classif* did not detect a shift.

No samples available as *Classif* did not detect a shift.

(e) Top different samples.

(f) Top similar samples.

Figure 49: Fashion MNIST randomized and original split, univariate two-sample tests + Bonferroni aggregation.

(a) Randomly shuffled dataset with same split proportions as original dataset.

(b) Original split.

Figure 50: Fashion MNIST randomized and original split, multivariate two-sample tests.

## A.2.3 CIFAR-10

(a) Randomly shuffled dataset with same split proportions as original dataset.

(b) Original split.

(c) Randomly shuffled dataset with same split proportions as original dataset.

(d) Original split.

No samples available as *Classif* did not detect a shift.

No samples available as *Classif* did not detect a shift.

(e) Top different samples.

(f) Top similar samples.

Figure 51: CIFAR-10 randomized and original split, univariate two-sample tests + Bonferroni aggregation.

(a) Randomly shuffled dataset with same split proportions as original dataset.

(b) Original split.

Figure 52: CIFAR-10 randomized and original split, multivariate two-sample tests.

## A.2.4 SVHN

(a) Randomly shuffled dataset with same split proportions as original dataset.

(b) Original split.

(c) Randomly shuffled dataset with same split proportions as original dataset.

(d) Original split.

(e) Top different samples.

(f) Top similar samples.

Figure 53: SVHN randomized and original split, univariate two-sample tests + Bonferroni aggregation.

(a) Randomly shuffled dataset with same split proportions as original dataset.

(b) Original split.

Figure 54: SVHN randomized and original split, multivariate two-sample tests.