[Reviews · NeurIPS 2019]

Reviewer 1



Originality: the paper focuses on rather classical question of distribution shift, with as few example as possible. It also proposes preliminary idea to identify examples that are representative of the shift,which seems to me more novel and beyond that, proposes to distinguish between benign and malign shifts. However this aspect is not very developped. The studt is restricted to "natural" shifts, ie. no attacks. Quality: the submission is similar to a review paper. Many technics are compared and very well presented, the supplementary material contains a lot of results that could be exploited. There is no theoretical part, which is not really a problem considering this paper. Section 2 is ok for the related work part, as far as I know. Section 3 is a (quite long) description of studied algorithms, followed by a (also quite long) description of used tests. Then comes 2 short parts on "most anomalous samples" and "malignency of the shift" : the main contribution of this work should be there, but the description is rather vague. I'm confident that the author could sumerize the way they select the most anomalous samples with more formalism (for instance : l 178 : "we can identifiy..." : there has to be an equation to make this think clearer?) Section 4 seems sufficient to reproduce results Section 5 : this part is more the experimental results summary than a discussion. I would expect in the discussion may be more intuitions or interpretations of the results. Clarity: There is a well-though labelling of different methods, such that one can read results more easily. The whole paper is very organized. Significance: the subject is interesting and there are good ideas in the paper. However the most significant part (characterization and malignency detection) are not supported enough to say this work can be used in its present form. Details, in the order of appearance in the paper: Section 3.1 : can you make clear, among used algorithm, which one are original (I suspect "label classifier" and "domain classifier" not to be standard tools?) or give a reference if they are not? l139 : you precise you used a squared exponential kernel : is it a usual setting, can it have an impact on results ? (the "we use" somehow indicates it's an arbitrary choice) l 165 166 : notation : c and r are not defined ll 196 : "the difference classifier" : which one is it? l 197 : I think a word is missing l219 : not sure of what "label proportion" refers to table 1 : how the "detection accuracy" is obtained? figure 2 /3 : I have a hard time trying to interpret all your figures (linked with previous comment on discussion part) : can you help your reader in this task? fig 3.c : I don't see why those examples are different? biblio : ref[3] was published in UAI 2018 ref [24] : typo "reviewpart" ref [38] : ICLR 2014?

Reviewer 2



UPDATE: Thanks for your reply and the clarifications about your work. In an updated version of the paper, I would recommend to compare against related work that does not follow the two-sample testing paradigm. --- Summary: The general framework for detecting dataset shift that is considered in this work consists of two main building blocks. The first one is a dimensionality reduction (i) followed by a two-sample test (ii) determining whether source and target data come from the same distribution. The authors consider a number of possibilities for both (i) and (ii). Here, the problem setting is that labeled data from the source and unlabeled data from the target domain is available. Moreover, the authors describe how to use a domain classifier to obtain the “most anomalous samples” to characterize the shift. Lastly, the paper proposes heuristics to determine how severe a shift is. Originality: The paper builds on a number of well-known techniques and combines them for detecting dataset shift. In that sense, the contribution is not very “original” but still an interesting analysis. I am not aware of previous work that has performed the same analysis. Quality: The authors themselves state that their ideas on characterizing the shift and determining its harmfulness are preliminary and heuristic. I agree with this judgement. In particular, I would like to see a discussion of the underlying implicit assumptions behind these ideas and the failure cases. For instance, in 3.4 1) the authors propose to look at the labeled top anomalous samples (i.e. those labeled examples the domain classifier predicts to be from the target domain with certainty) and use the black-box model’s accuracy on those as a characterization of the shift. Assuming such labeled anomalous samples exist is of course a pretty strong assumption. Related to that point, the authors state in line 196: “Since the difference classifier picks up on the most noticeable deviations, the domain classifier’s accuracy functions as a worst-case bound on the model’s accuracy under the identified shift type.” I don’t think this statement is true: if the domain classifier’s accuracy is large would mean that the shift is large and hence the (source domain) model’s accuracy is probably rather low. Could you please clarify the statement? Regarding the empirical study, the comparison between the different shift detection techniques seems thorough but I wonder why there is no related work or other baselines in the experimental results section. I would also recommend to comment on how the distribution of the p-values changes as the number of samples from the test set increases. Can you characterize in which cases they decrease monotonically with the number of samples and when they do not? Clarity: The paper is clearly written for most parts but I found some of the experimental details hard to understand. In particular, the results in Table 1 seems to be averaged over MNIST and CIFAR-10 which I find a bit odd. Also, it is not clear to me what sort of intervals are shown in the plots -- is that information given somewhere? Significance: I do find the considered problem setting important and more work is needed in this area. However, the main doubt I have about this paper is that related work is not considered in the experiments, the weaknesses of the approach are discussed too little and the ideas on characterizing the shift and determining the harmfulness are too preliminary.

Reviewer 3



The authors conduct quite a fairly detailed study of different ways of thinking about statistical tests for detecting distribution shift. The paper and contribution is well-contextualized and the manuscript is clear. There is a clear takeaway message for engineers designing tools for monitoring deployed machine learning models to use pre-trained classifiers with single-dimensional statistical testing. The shortcoming, which the authors themselves bring up is that there is no temporal dependence considered (i.e. streaming data) and that only images are considered. To be really comprehensive, data of these other forms could have been considered.

[Author Response · NeurIPS 2019]

**General Reply**

We thank all reviewers for their constructive comments. Overall, the reviewers appreciated the importance of the problem, the practical utility of our results, the rigorous experiments, and the clarity of our writing. At the same time we recognize that the scores were split, with R3 advocating acceptance and R1, R2 leaning towards weak reject. We are optimistic that our rebuttal can address each reviewer's individual concerns below:

**Reply to R1:**

Thanks for your thoughtful comments on how we can improve our draft.

***"The stud[y] is restricted to "natural" shifts, ie. no attacks."*** In fact we do run experiments in which shifts are due to adversarial attack, finding, interestingly that here two-sample tests performed in the ambient space are most effective.

***"the submission is similar to a review paper."*** While we agree that our work draws upon many other papers (like a review) would, our contribution here is a rigorous empirical study with results that stem from combining existing techniques in new ways. BBSD has never been applied beyond the label shift setting and its wider applicability (in practice) had never previously been demonstrated. Moreover, the general DR + 2-sample test pipeline has never been systematically investigated. We believe that this constitutes a source of fundamental scientific insights. In short, we wish to assert that the novelty of our paper is in the science, not the introduction of new models.

***"the main contribution of this work should be there,"*** While we agree that identifying benign vs malign examples is of paramount importance, we hope to convince you that detection itself, which is better developed in our work is a sufficiently vital problem. We hope to make more progress on detecting malign examples in future work.

***"the author could [summarize] the way they select the most anomalous samples"*** While we considered two approaches, our promising results to date consist of selecting those examples that a domain classifier assigns with highest probability to the target domain.

***"Better interpretation of results"*** We will discuss our intuitions and interpret the results more thoroughly if accepted.

***Details:*** *a*) **Sec. 3.1**: We will clarify that all of our DR + two-sampling methods are non-conventional. The label classifier applied broadly as a shift detector is the most successful technique and indeed our contribution. However, domain classifier approaches have been explored previously. *b*) **L 139**: We chose the Gaussian kernel for MMD following the original paper. Per your suggestion, we will add ablations to show sensitivity to kernel bandwidth. *c*) **L 165-166**: $c$ and $r$ refer to the number of columns and rows of the contingency table. *d*) **L 198**: "difference" classifier refers to the domain classifier (pardon the typo). *e*) **Tab. 1**: Detection accuracy measures how often, across a variety of shifts, a statistically-valid detector identifies the shift. *f*) **Fig. 3**: For this dataset, differences between source and target data arise from a different set of angles from which the items were photographed (source: $0° - 175°$, target: $180° - 355°$). *g*) **Typos and reference bugs**: Thanks for the attention to detail. We will make all appropriate changes in the camera-ready if accepted.

**Reply to R2:**

*a*) **L 196**: You are right. This statement entered the paper due to a miscommunication among collaborators. We apologize for the error and will strike it from the camera-ready version if accepted. You are also right that this heuristic is predicated on strong assumptions (addressed below). *b*) **Access to labeled anomalous samples is a strong assumption**: Our intuition here is that while labeling all examples at deployment would be prohibitive but that a small quality team might be used to selectively label those *top anomalous* examples. Such monitoring teams are increasingly common (Google's search quality team is a famous example). *c*) **Related work and other baselines**: We point out that all standard two-sample tests in the input space constitute baselines as does the domain classifier approach. Notably (Tab. 1) MMD on inputs performs worst among all tested methods. *d*) **Distribution of p-values in experiments**: This is a great question. We will investigate the distribution of p-values and add these results to the paper. *e*) **Why is Tab. 1 averaged over MNIST and CIFAR-10**: In this table, we average over all different perturbation models and datasets because our goals was to see if there exists a shift detection method that consistently outperforms the others. We will make this rationale clearer. *f*) **What sort of intervals are shown in the plots**: the intervals around the (mean) p-value are 1-$\sigma$ error bars. See L 222 - 225 and the caption of Fig. 2 for details. Time steps used for testing are described in L 229. *g*) **Weaknesses, underlying assumptions and failure cases w.r.t. shift characterization and malignancy detection**: The big assumption here is that *obviously* shifted examples are somehow more likely to be misclassified. This assumption might not hold. We will work to assess the correlation of domain classifier confidence with misclassification in real datasets and to formalize settings in which the assumption is reasonable.

**Reply to R3:**

Thanks for your strong positive review for our paper. We are glad that you appreciated the usefulness of the approach and the lack of pretense. We agree that further experiments with data from other domains and extensions to handle streaming data are promising next steps.

[Meta-Review · NeurIPS 2019]

The reviewers all considered the author feedback and discussed the paper thoroughly. Most concerns could thereby be clarified and the main point against accepting the paper is a lack of comparison with approaches not based on two-sample tests like [17,19]. If these could be included in a camera-ready-copy, that would be great; but even without it, I consider the contribution sufficient to justify accepting the paper as a poster.